# hGLUTEN Tool: Measuring Its Social Impact Indicators

**DOI:** 10.3390/ijerph182312722

**Published:** 2021-12-02

**Authors:** Antonia Moreno, Guillermo Sanz, Begonya Garcia-Zapirain

**Affiliations:** eVIDA Research Group, University of Deusto, Avd. de las Universidades, 24, 48007 Bilbao, Spain; guillermo.sanz@opendeusto.es (G.S.); mbgarciazapi@deusto.es (B.G.-Z.)

**Keywords:** social impact, artificial intelligence, embedded system, sensor NIR, ODS

## Abstract

hGLUTEN is a technological solution capable of detecting gluten and spoiled food. We measured the social impact of the hGLUTEN tool using two Likert scale surveys with two groups: professionals (engineers/chefs) and end-users. These data have been assessed in accordance with the social impact indicators defined for the Key Impact Pathways introduced by the European Commission for Horizon Europe and the criteria of the Social Impact Open Repository (SIOR). A total of 85% of users, 100% of engineers and 68% of professional chefs consider it very relevant to participate and give their opinion in research projects, which shows the increasingly high level of involvement of the general population. A total of 88% of users were unaware of other applications that detect gluten and were more dependent on guidelines provided by allergy associations and expiry dates of foodstuffs. In addition, only 5% of professional chefs said they were aware of other technology capable of detecting gluten in food, which may indicate a large economic market and good commercialisation possibilities for the tool in the future. Finally, the inclusion of tools to motivate users to promote it has been identified as an area for improvement, which could mean that it should be made more visible in the media to increase its impact and influence.

## 1. Introduction

Food allergies and intolerances affect an increasing number of people. For this reason, the eVIDA research group has developed a technological solution called hGLUTEN (Figure 1), which detects the presence of gluten and spoiled food with an accuracy of 88% and 100%, respectively. The tool offers real-time measurements without having to send samples to the laboratory, using artificial intelligence or NIR spectroscopy. By modifying the analysis process, the use of this easy, fast, reliable and low-cost method would reduce costs for the user, the food industry and the health sector. This article analyzes the social impact that this tool could have, following the indicators established by the European Commission.

Food security has evolved over the years. The most comprehensive understanding of food security is continuously having enough safe, healthful food at an affordable cost to meet nutritional and caloric needs and the preferences of the individual or household [1].

Food allergy is known to negatively impact the psychosocial aspects of the health-related quality of life (HRQL) [2,3,4,5,6,7]. It affects nearly 5% of adults and 8% of children, with evidence of an increasing prevalence. It concerns not only individuals but also their families, with higher levels of stress and anxiety described in families living with the risk of anaphylaxis [8,9,10]. QoL is described as individuals’ perception of their position in life in the context of their culture and values, in relation to their goals, expectations, standards and concerns [11]. Whilst patients may carry the diagnosis of food allergy into adulthood, Kamdar et al. have recently reported that at least 15% of patients show adult onset develop of food allergies [12].

Food ingredient labels must be read every time before food is served, and simple tasks such as grocery shopping become time-consuming and often expensive endeavours for families who have a member with a food allergy [13]. Despite the best attempts of families to maintain a safe diet, accidental ingestions of food have been reported to occur in about half of these patients [14]. As there is no cure, the management of the condition consists of strict avoidance of allergen exposure and prompt treatment with epinephrine in the event of an allergic reaction [15].

Providing information about the presence of an allergen in a food is an important tool for risk management, as consumers with food allergies often rely on labelling (i.e., ingredients list; advisory and precautionary labelling) to assess the safety of pre-packed foods [16,17]. This information is used in conjunction with other risk management strategies, such as reliance on past experiences of consuming a food product, sensory appreciation of risk and assessment of product qualities [18]. 

On 1 April 2016 the UN General Assembly declared the Decade of Action on Nutrition programme, which would run from 2016 to 2025, as part of the UN Sustainable Development Goals initiative [19]. As the world moves towards achieving the Sustainable Development Goals by 2030, goals 2 (end hunger), 3 (improve health), 8 (decent work and economic growth), 12 (responsible consumption and production), 13 (climate action), 14 (life below water) and 15 (life on land), are all deeply interlinked with the global food system [20]. 

Food is considered a basic human right—as echoed by the United Nations Declaration of Human Rights, “people have a right to freedom from hunger, and everyone has a right to have access to adequate food” [21,22]. A right to food is a claim that is accepted and acknowledged by individuals and all of society, without which individuals would not survive or maintain their well-being [23].

Access to healthy food, which is adapted to each individual’s needs, is increasingly sought after by society as a whole and is practised at home. From that perspective, while more frequent home cooking is associated with better diet quality and is generally perceived to be healthier and more affordable, low-income and food insecure households still face multiple barriers to preparing healthy meals [24].

In this sense, people now demand higher quality products, ranging from the latest technologies to the most widely consumed goods in supermarkets. However, product quality is not only a consumer concern. Producers are equally interested in high quality to ensure their position on the market or their reputation, which are vital to their business profits.

Spectroscopy has been an alternative to traditional chemical and biochemical techniques since the 1960s at the international level, demonstrating good potential to obtain accurate and swift assessments of the chemical composition of some biological systems. The interaction of luminescence with atoms is controlled by measuring the reflectance within the visible (VIS) and infrared (IR) spectrum.

The purpose of this project was to carry out a technique capable of examining the properties of solid and liquid foods and the allergens they contain by studying absorption and reflection of the different wavelengths of the foods in the near infrared spectrum range between 900 and 1700 nm. Other sensors will be used to analyse the quality of foods and food spoilage.

The hGLUTEN project provides a technology solution that can analyse food fingerprints using absorbance, reflectance and sample signal values. It also employs an artificial intelligence algorithm capable of detecting gluten-free foods that contain less than 20 ppm of gluten as well as some spoiled foods. The system design includes a near-infrared spectrometer, a pH meter and an infrared temperature sensor, which we use to monitor milk lactic acid fermentation and the consequent pH reduction. All of the above form an integrated technology solution, using a Raspberry Pi. By pressing a button, we began the test and obtain the results on a web server or an app. These tests reduce sampling time and the amount of material used in the laboratory. A total of seven gluten-free samples, six samples containing gluten, five types of milk and four types of fruit were validated with the use of artificial intelligence, reaching 88% accuracy in the detection of gluten allergy, together with laboratory instruments such as a digital microscope or reactive tests. We hope to enlarge this database in the future and carry out an analyser for other allergens such as lactose or nut traces.

### 1.1. Social Impact of Research

The social impact of a project takes place when the scientific knowledge has been produced, published and transferred to society and its institutions, prompting positive effects in them. However, this social impact can be analysed from various perspectives, rather than only the economic one. It is people, society as a whole, who know their needs and how social impact can satisfy such needs [25].

The way in which research is organised and funded in every scientific domain is subject to growing scrutiny. Economic investment in health research is used by policymakers, health care providers and doctors to make key decisions based on evidence pursuing the patient’s benefit. They also focus on ensuring that limited medical resources are used to the best advantage to provide effective and sustainable services [26].

Governments, funding agencies and research organisations across the world increasingly seek to maximise social and economic returns from research investments, shaping policies and research practices. For instance, in the European Union’s Horizon 2020 research and innovation programme, excellent science, industrial leadership and societal challenges are three mutually reinforcing priorities [27].

The pathways to identifying how scientific research impacts society and improves policy and practice are diverse. Some highlight the need to focus on how a research project generates productive interactions, which are understood to be exchanges between researchers and stakeholders that constitute the framework wherein robust scientific knowledge is produced and assessed [28]. Permanent dialogue between researchers and non-researchers is strengthened in these interactions from the start of the research to the final presentation of the findings [28]. 

We therefore believe that it is important and necessary to measure, in the short term, the social impact of this research, which makes use of the hGLUTEN tool to analyse the presence of gluten in food and detect possible food spoilage. It is also necessary to improve the mechanisms that evaluate the impact that this scientific research is already producing. Accordingly, with a view of providing a permanent system of indicators for impact evaluation and contributing towards ensuring the social impact of research, the European Commission selected the FP7 EU Project IMPACT-EV. This project designed the first global open-access repository on science social impact called SIOR (Social Impact Open Repository). Van den Besselaar, Flecha and Radauer (2018) [29] elaborated a set of indicators, sources and methodologies to measure social impact in the short, medium and long term for each of the Key Impact Pathways, which we will follow in this study.

### 1.2. hGLUTEN Technological Solution Description

The project consists of different technological parts that will be described in the following subsections.

#### 1.2.1. Absorption/Transmission/Reflection Spectroscopy

Ultraviolet–visible–near-IR spectroscopy (UV-Vis-NIR) is used to measure the absorption, transmission and reflectivity of a variety of technologically important materials such as pigments, coating, windows and filters. This more qualitative application generally requires registering at least part of the spectrum to specify the optic or electronic properties of the materials. 

Transmission spectroscopy is closely related to absorption spectroscopy. This technique is used for sampling solid, liquid and gas. Here, light passes through the sample and is compared with light that does not. The resulting spectrum depends on the length or thickness of the sample’s path, absorption coefficient and reflectivity, the incidence angle, the polarisation of incident radiation and, in the case of the particles, their size and orientation. In the Beer–Lambert law (Equation (1)), the term IT/Io is called transmittance. The configuration of this form of spectroscopy is similar to the one used for absorption.
*Einnt* = *Eel* + *Evib* + *Erot*
(1)

*Eint* = Internal energy

*Eel* = Electronic energy

*Evib* = Vibrational energy

*Erot* = Rotational energy

Reflectance spectroscopy is the study of light based on the wavelength that has reflected or dispersed in a solid, liquid or gas. When photons enter a mineral, some are reflected off grain surfaces, and others pass through the grain or are absorbed. Photons that are reflected off grain surfaces or are refracted through a particle are said to be scattered. Scattered photons may find another grain or scatter off the surface so that they can be detected and measured. All materials have a complex refractive index (Equation (2)):*m* = *n* − *jK*
(2)

*m* = complex index of refraction

*n* = real part of the index

*j* = (−1) × 1/2 *K* is the imaginary part of the refractive index, sometimes called the extinction coefficient.

#### 1.2.2. DLP NIR Nano Scan by Texas Instruments 

Spectral signatures in the 780–2500 nm range are dominated by the presence of hydrogen bonds such as OH, CH, Nh and SH. The NIR frequency band is therefore particularly suitable for use in food and agricultural control and health diagnoses as well as the petrochemical processing and pharmaceutical industries.

Within the NIR frequency band, each spectroscopic application has unique requirements in terms of wavelength and chemometric analysis. For example, the DLP NanoscanNIR (900–1700 nm) can provide information on water (H20) and sucrose (C12H24O12) content. Extending the instrument’s wavelength range to 2500 nm enables it to discover additional organic compound signatures and may improve results in pharmaceutical process supervision.

The DLP NIR Nano Scan (Figure 2) is a compact battery-operated spectrometer evaluation module for portable near-infrared solutions. With the digital micromirror device (DMD) DLP2010NIR, the NIRscan Nano supports Bluetooth low energy to enable mobile lab measurements for hand-held spectrometers.

The EVM incorporates the DLP2010NIR DMD, a diffraction grating, and a single element detector to replace expensive GaAs linear array-based detector design. With its TI Tiva™ TM4C1297NCZAD processor, databases in the cloud can be leveraged through cellular networks for real-time lab equivalent analytics, which allow food or skin analysis and wearable health monitor solutions. Developers can also create their own data collection and analysis through innovative iOS and Android applications. 

#### 1.2.3. Block Diagram of the Proposed Solution 

As regards the system architecture (Figure 3), a Raspberry PI 4B microprocessor was used in order to link all the sensors and integrate it with the system to connect to the web server or app to obtain the results. 

The following sensors were used:pH meter: to measure the acidity level of foods and analyse its change in them;NIR spectrometer to measure the gluten level in foods;Temperature sensor: to observe how temperature can affect changes in the condition of foods.

## 2. Methods

To explore and measure the social impact that the hGLUTEN tool has had, an online survey was conducted using a Likert scale with two groups: professionals (engineers/chefs) and end-users. This data has been assessed in accordance with the social impact indicators defined for the Key Impact Pathways introduced by the European Commission for Horizon Europe, and the criteria of the Social Impact Open Repository (SIOR). In the following we will explain the method used, the data collection, the description of the participants and the data analysis.

### 2.1. Data Collection

Data collection took place through an online Likert-type scale survey in which eleven questions were asked to professional cooks and engineers (Q11P), and another 14 questions were asked to users of the hGLUTEN tool (Q14U). The survey was active for a period of two months from 23 February 2020 at the link (https://forms.gle/xobpmyuvkTRV82pr9). An invitation to participate in the survey was sent by email to students and professors of the Faculty of Engineering of the University of Deusto (Engineers) and the School of Hostelry Management of Leioa at the University of the Basque Country (Professional Chefs). In the case of standard users, an e-mail was also sent to the university community of Deusto. Together with these mails, the user’s guide on how the hGLUTEN tool works and an infographic of its design were attached. In order to carry out the calculus operations, Microsoft Excel was used. The survey questions were grouped into eight social impact indicators:

From the point of view of professional chefs and engineers, the indicators include:(1)Change in behaviour and practices;(2)Change in the improvement of work practices;(3)Evaluation and impact on professionals;(4)Satisfaction with follow-up and treatment.

From the users’ point of view, the indicators include:(5)Social improvement, health and quality of life;(6)Improvements in the person’s experience;(7)Improvement in the environment;(8)Improvement in services provided by professional chefs for better monitoring and treatment of pathology.

#### Description of Interviewees

A total of 62 respondents participated in this survey: 17 engineers, 19 professional chefs and 26 standard users, of whom were 35 men and 27 women. Participants ranged in age from 18 to 60 years old. For the analysis of the impact indicators of the hGLUTEN tool, we considered on the one hand the co-creation process involved in its technological development, in which engineers and professional chefs participated. On the other hand, we included the more general vision of the standard users, although not all of them were celiacs or had gluten intolerance.

Although the non-response rate was high, the survey can be regarded as representative at least of an academic environment. For all three groups the results can be regarded as significant with a confidence level of 90% and an error margin below 20%. The specific values are as follows:

Standard users: error margin 16.71%, confidence level 90%, 26 of 15992 responded;

Engineers: error margin 19.91%, confidence level 90%, 17 of 1580 responded;

Professional chefs: error margin 18.63%, confidence level 90%, 19 of 573 responded.

As interviewees tend to take part in surveys which they regard as important and represent their own issues or interests, the survey is likely to contain a bias, but on the other hand participants tend to choose midpoint while filling forms [30,31,32]. 

### 2.2. Data Analysis Method

#### Categorical Data

This survey was divided into two different surveys according to the profiles which they focused on: the first one was answered by engineers and professional chefs and the second one was answered by standard users without any specified occupation.

All questions of the survey can be classified by the answer possibilities, e.g., yes/no questions and multiple-choice questions. The multiple-choice questions have two different 1 of n answer scales. The first one (Ch1) contains answers on a scale beginning with a zero value, e.g., never, rarely, sometimes, often and always, while the answers of the second type (Ch2) have a central value with 4 symmetrically placed relations, e.g., totally disagree, disagree, neither agree nor disagree, agree and totally agree.

All answers were encoded as follows, and yes and no were replaced by 1 and 0, respectively. Labels from scales beginning with a zero value were encoded as 0 to 4, and labels from symmetric scales were encoded as −2 to 2. All calculus operations were performed with this encoding. The descriptive results are noted in Table 1 for Engineers/Professionals chefs and Table 2 for standard users The nearest number to the mean can be decoded with the above explanation and interpreted as a general answer to the question with a tendency to the next (resp. previous) value on the scale if it was rounded up (resp. rounded down).

### 2.3. Social Impact Analysis

Van den Besselaar, Flecha and Radauer (2018) established the social impact indicators that were collected in the Monitoring the Impact of EU Framework Programmes [29] and that we will use for our research. For each Key Impact Pathway, indicators are defined that measure the expected progress in social impact and that we will take into account for the present study. For the purpose of our study, the data collected were analysed under this framework, which includes the four Key Impact Pathways: (a) achieving R&I missions, (b) addressing global challenges, (c) engaging EU citizens and (d) supporting policymaking [29]. Consequentially, this paper reports quantitative and qualitative data as a result of measuring the social impact of the hGLUTEN tool.

## 3. Results and Discussion

Although the research impact assessment (RIA) has gained recent interest, it is not new [27,33,34,35,36,37,38]. RIA uses a multitude of methods from social science disciplines to examine the research process with to the aim of maximising its societal and economic impacts such as intellectual property, spin-out companies, health outcomes, public understanding and acceptance, policymaking, sustainable development, social cohesion, gender equity, cultural enrichment and other benefits [27].

The creation of the Social Impact Open Repository (SIOR) was based on the concept of social impact defined by the European Commission, which refers to economic impact, social impact, environmental impact and, in addition, impact on human rights. SIOR is also the world’s first open-access repository [39,40,41]. Social and policy impact repositories are likely to become fundamental tools for the evaluation of social and policy impact, such as journal rankings in the Scopus or Web of Science currently are for the evaluation of scientific impact [27]. The idea of impact and its measurement are important tools to formulate and evaluate science, technology and innovation policies. As can be seen, we grouped the results obtained in the online survey into eight social impact indicators.

In order to carry out the evaluation to measure the progress of the social impact of the hGLUTEN tool, the indicators defined by the European Commission were taken into account, in which the target groups of the research are decisive. In this case, a standard user was considered to determine whether the use of the designed technology would be useful for him/her and professional chefs, who could use it on a daily basis in their jobs (restaurants, catering, hotels, etc.). Progress is currently being made on improving the tool with the Euskadi-EZE Coeliac Association. The vision that the engineers would have when developing this technology would also be important in assessing its social need and technological possibilities.

### 3.1. Social Impact and Its Measurement through an Online Survey

Below we show the social impact indicators, which are derived both from the online survey of engineers and chefs on the one hand and standard users on the other. 

#### 3.1.1. From the Point of View of Professional Chefs and Engineers

-Indicators of change in behaviour and practice include:

Q1P: The new application would help to avoid anaphylactic incidents in people allergic to gluten or spoiled food. 

Q2P: Authorities could improve their policies and programmes with the data that the app provides. 

Q3P: This information has been shared with family, friends, etc.

As can be observed in Table 3, most of the engineers and professional chefs agreed (41% and 58%) or totally agreed (30% and 21%), respectively, that developing this tool could lead to the avoidance anaphylactic incidents due to the ingestion of gluten or other problems caused by the consumption of spoiled food. Furthermore, many of them believe (engineers, 59% and professional chefs, 53%) that its use could bring about changes in health programmes to encourage prevention policies. Nevertheless, 18% of the engineers have not shared information or remarked to other people about this tool, 24% have shared little information, and 5% professional chefs have not shared any information, while 26% have shared little information. This could mean that the message of sharing the experience with other users or with the media is necessary to improve its impact. In general terms, these percentages indicate the positive perspective about the usefulness of hGLUTEN and prove the need for it, as well as concern about these food-related problems. 

-Change indicators in the improvement in work practices include:

Q4P: The use of this tool could lead to changes in work practices;

Q5P: It would improve the detection of allergens and enable more personalised treatment than with other tools that people knew or used;

It may seem contradictory that professional chefs (53%) and engineers (82%) think that this tool would not result in any changes in their work although, as previously mentioned, they have a positive opinion of the tools. This may be explained by their failure to understand its use protocol and therefore, it would be advisable to find a way to demonstrate this to them. It is also worth mentioning that the reasons for which 47% of professional chefs would note changes in their work would range from their being capable of creating new dishes to providing their customers with greater convenience, security and service. It would also enable decision making on whether to use certain ingredients. All of the above could lead to a more personalised approach to the detection of other allergens, which the respondents particularly agree with (engineers 65% and professional chefs 42%) or totally agree with (29% and 21%, respectively) (Figure 4).

-Indicators of evaluation and impact on professionals include:

Q6P: I knew about some type of scientific tool to enable people to detect allergens (gluten) or spoiled foods. 

Q7P: The tool should be replicated for the detection of other types of non-gluten allergens.

Q8P: We consider it necessary and important to give our opinion on research processes such as this one.

A large majority of the professionals surveyed are in favour of replicating the tool to include detection of other allergens. A total of 53% of engineers totally agree and 41% agree, and 26% of professional chefs totally agree (Figure 5). Both engineers (100%) and professional chefs (68%) believe it is important to give their opinion in research projects for various reasons. Some of the reasons stated were as follows: (a) to gain knowledge and experience to share (b) raise awareness that science and technology should advance to help society (c) the need to gather more information from different viewpoints that would improve the app (d) the suitability of promoting feedback on societal needs e) interest in knowing that this research is being conducted. Only 5% of the professional chefs stated that they knew another technology capable of detecting gluten in food, which may indicate there would be a large market and good possibilities for commercialising the hGLUTEN tool in the future.

-Satisfaction indicators of follow-up and treatment include:

Q9P: Compared to other existing options, this new application could be used to customise data collection and processing.

Q10P: The hGLUTEN tool could be more involved in the treatment process.

Q11P: With this app, I could do a better job monitoring my customers’ allergy problems related to allergens and spoiled food. 

This tool could improve data gathering and personalised treatment for people who suffer from allergies to gluten or spoiled food. The majority of engineers (71%) and professional chefs (63%) agree. It is also noteworthy that hGLUTEN would give them a bigger role in these persons’ treatment (82% of engineers agree, 63% of professional chefs agree), and therefore, it would be possible to monitor consumers’ food allergies more closely (53% of engineers agree, 41% totally agree and 68% of professional chefs agree and 11% totally agree) (Table 4). Professionals think that this is a “simple, fast, effective, easy to carry app that anyone could use, which would make it possible to reach a large number of users and that it is also capable of creating alerts”. In addition, the following remarks were made: “as professionals lacking knowledge of other tools that can detect gluten, this is a good option” and “it could improve the quality of the diet of risk groups” and “its use could be applied on a large scale in food companies; compiling a great deal of information in very little time and without the need for chemical treatments” indicate that this is a very saleable technology. 

#### 3.1.2. From the Users’ Point of View

-Indicators of social improvement, health and quality of life include:

Q1U: Patients with pathologies caused by allergy to gluten, allergic reactions to spoiled food and even healthy people should have easy access to such services.

Q2U: This app would be effective for treating people with allergies to gluten or spoiled food. 

Q3U: This app would prevent possible complications in pathologies caused by allergens that we are exposed to. 

Q4U: In research processes such as this one, I consider it important to give my opinion.

As can be observed in Table 5, possible users of this app totally agree (73%) that people with pathologies caused by allergy to gluten and those who are allergic to spoiled food should have easy access to the tool. This would substantially improve their quality of life as well as making their treatment more efficient (58%). Users are also very aware (42% totally agree and 31% agree) that the application could prevent the complications caused by risk situations. Therefore, 85% of these respondents clearly think that it is vital to give their opinion as potential hGLUTEN users. Some of the reasons they stated were (a) the user is the ultimate recipient of the product and could influence the co-creation process, (b) to provide an external point of view to the research team, (c) their help can make the app easier to use, (d) because they have family members with food allergies and (e) due to the lack of information on how to obtain gluten-free food or simply to help science advance. All of the above show the general population’s high level of involvement. 

-Indicators of improvements in the person’s experience:

Q5U: Access to data processing and the usefulness of tool portability. 

Q6U: I would have instant access to the data whenever I needed it during the processing process.

Q7U: The tool’s interface must be easy to use to be effective.

Q8U: I know about another tool that people use to measure allergens in food or detect spoiled food. 

Q9U: With hGLUTEN I could make as many measurements as I wanted or needed.

The users surveyed stated (Table 6) that they found the easy portability of the tool very useful (38%) or quite useful (38%). It gives access to information on gluten or spoiled food anywhere very quickly, which the users deemed very important, always (38%) or often (38%). Based on the user guide, the respondents found that the application interface is simple to use, and only 4% did not agree. In addition, 88% did not know other applications that detect gluten, and they mainly depended on information supplied by allergy organisations. In relation to the detection of spoiled food, they stated that they read expiry dates. Lastly, it is important to highlight that users can take as many measurements as they wish, which enhances the user’s experience, always (39%), often (42%) and sometimes (19%).

-Indicators of improvements in the environment include:

Q10U: This new app would increase the quality of life of people allergic to gluten and improve the service they receive in cafés, restaurants, schools, hospitals, etc. for example, by reducing travel time, leaving them more available time, using other alternative technologies or ways to save on other tools or treatments. 

In this indicator the users seem to have a high level of awareness of environmental issues and the energy savings that the use of this new tool can provide, with much awareness 65%, a lot of awareness (23%) and not much or less awareness (12%). It seems evident that health costs could be reduced through hospitalisation prevention with the tool; fewer health care workers would be devoted to monitoring and treating these patients, and savings on prescriptions, lower costs for future industrial food production and shorter working times for staff concerned could be achieved.

-Indicators of improvements in services provided by professional chefs for better monitoring and treatment of the pathology:

Q11U: Food services staff would monitor and handle risk situations more easily with this app.

Q12U: In order to understand how hGLUTEN works, the data provided by the engineers is important.

Q13U: I have commented and shared with others (friends, family, etc.) the knowledge and advantages that this application provides. 

Q14U: If there was no medical or family support, the adaptation of this application would be good for the person using it in terms of complexity and knowledge of their disease.

The users who responded to the survey believe that use of the app by professional chefs would mean better control of risk situations related to the food they serve. In this respect, it should be noted that nobody disagreed or totally disagreed (Table 7), which again implies a positive opinion of the tool’s use and efficacy. In turn, only 8% stated that the user guide with technical information on the tool provided by the engineers was little relevant or nothing relevant (0%). This confirms that it is a user-friendly app. It also appears that it is easy to use hGLUTEN (62% agree and 38% totally agree) without the help of doctors or family members, as it is not complex. Some respondents have not shared knowledge of this tool with others (19% have shared nothing about the app, and 15% have shared little information). This may mean that it should be made more visible in the media so as to extend its use and improve its impact.

### 3.2. Social Impact Assessment 

Based on the results obtained from the online survey on the use of the hGLUTEN tool by end-users, professional chefs and engineers, social impact indicators have been applied to measure the progress related to the three Key Impact Pathways (1) “Addressing global challenges” (2) “Achieving R&I missions” and (3) “Engaging EU Citizens”. 

#### 3.2.1. Addressing Global Challenges

The use of the term Social Impact is limited only to those cases in which the application of scientific results in a given social reality achieves its social improvement; that is to say, when a society reaches some of the social goals [42]. In this case, the SDGs: 3—Health and well-being, 9—Industry, innovation and infrastructure, 10—Reducing inequality, and 11—Sustainable cities and communities. This contribution covers some of the global challenges according to the short-, medium- and long-term indicators which have been identified:Short-term indicators: (a) The creation of a database which contains personalised information on each patient’s possible exposure to allergens and spoiled food and predictability of risk situations by using artificial intelligence (SDG: 3); (b) Increased knowledge among engineers, professional chefs and end users about when food allergens affects us personally (SGDs: 3, 9, 10 and 11) and recording said information in alarm moments; and (c) High replicability, as it can be adapted to detect any type of allergen (SDGs: 3, 9 10 and 11).Medium-term indicators: (a) Technology transfer and adoption of new knowledge: a system to methodically inform on the number of changes in professional chefs’ protocols, practices and work; (b) Technology transfer and the adoption of new knowledge for users/consumers, who will have alarms to warn them of immediate risks; (c) Technology transfer in engineering studies, other research groups, restaurants and institutional bodies that can include and improve this knowledge when handling information on gluten or spoiled food and how they can affect citizens (SDGs: 3, 7, 11); (d) Information that is easy for users/consumers to monitor and access with their mobile telephones (SDGs: 3 and 9); and (e) Foreseeing changes in chefs’ protocols and work practices: early detection and innovation with use of other products to replace those that contain gluten or evidence spoilage (SDGs: 9 and 11).Long-term indicators: (a) Social impact: the number of qualitative and quantitative tests on improved techniques and assistance as a result of knowledge transfer related to gluten content and spoiled food and the consequences for users, which will be carried out by research groups, restaurants, or food companies interested in new projects; (b) The number of improvements regarding previous situations and protocols for users who are allergic to gluten or may consume spoiled foods in different places: the home, restaurants, dining rooms in schools, hospitals, etc. (SDGs: 3, 9, 10 and 11); (c) The possible replicability and sustainability of these improvements by manufacturing new portable devices, using other recognition sensors for different allergens and conducting other more personalised tests and improve portability; and (d) There is emerging evidence of potential and real social impact on social media networks. The results indicate that scientists need to include the aim of sharing the social impact of their results in their dissemination and communication endeavours [43]. Check the social impact of the research on social media sites Twitter, Facebook, The Conversation, number of articles downloaded from databases, number of citations, etc.

#### 3.2.2. Achieving R&I Missions

A better understanding of how scientific and technological knowledge development affects society will open new avenues for reflection on strengthening beneficial changes or correcting unfavourable ones. This enables assessment of the pertinence of funding science and innovation [44]. The use of the hGLUTEN tool is in line with the approach of European research and innovation missions, whose objective is to provide solutions to issues that are socially relevant, for instance, the capacity to improve health, nutrition or living environments [45]. The contribution of hGLUTEN to achieving some of these missions in the short, medium and long term has been assessed as follows:Short-term indicators: (a) hGLUTEN is a tool based on knowledge transfer and the adoption of new knowledge that arose from the HealthyAIR project, which detects pollution risk in real life. In this sense, funds have been obtained from the Elkartek call of the Basque Government in Spain with the TECAM project to develop another tool to detect the presence of pathogens in dairy and gluten and soy intolerance with cross-validation of NIR to traditional PCR.Medium-term indicators: (a) Registration of the hGLUTEN utility model to resolve the legal aspects of its technological development; and (b) In the future, the information can also be stored in the cloud and be more accessible to patients, doctors and professionals who work as cooks in restaurants, hotels, school canteens, etc.Long-term indicators: (a) Development of new funded research to detect other types of allergens found in food, achieved thanks to the success and usefulness of HealthyAIR, which was the project that preceded the idea on which hGLUTEN was based; and (b) An increase in the number of impacts produced in the society, be it celiac associations, hotels, restaurants, etc., contacted to develop the tool and improve their quality of life. This will be achieved with the increased number of funded projects and the increased number of end users reached as a result of the expansion of hGLUTEN in the detection of other allergens.

#### 3.2.3. Engaging EU Citizens

In view of European states’ difficulties to meet the social demands of goods and services, there is a growing need for society, together with the concept of social innovation, to play a leading role [46]. So far, food samples have had to be taken to a laboratory to detect the presence of allergens in them, but with the hGLUTEN tool, this can be done in real time. We are currently working locally (Celiac Association of Euskadi-EZE and Leartiker S.Coop. Technology Center), but we are in talks with different partners in Spain and Europe to submit to calls that would fit this research, such as: HORIZON-HLTH-2022-DISEASE-03-01: European partnership fostering a European Research Area (ERA) for health research or HORIZON-HLTH-2022-IND-13-04: Setting up a European Smart Health Innovation. In the case of testing the hGLUTEN tool, participants are engaged, and progress has been assessed in the short, medium and long term:Short-term indicators: (a) Users and professionals (engineers and professional chefs) have participated in the process and been involved in the co-creation of different activities related to health and wellbeing, innovation, reducing inequality and achieving sustainability of cities. All the participants, engineers, users and chefs received the user guide, which explains how the tool works. They were later given an online survey on its ease of use and social impact indicators. This co-creation process is necessary to improve the tool and pursue its future replicability.Medium-term indicators: four types of publications or events were held to boost citizen participation in the scientific tests to improve the projection of the initiative and promote detection of allergens and spoiled food.Long-term indicators: (a) Dissemination through eight types of publications focusing on boosting citizen participation in the creation and use of scientific tests to improve projection and promote the detection of gluten and spoiled food; and (b) A measurement of citizen participation on social media sites regarding activities, programmes, products and innovation based on the project.

### 3.3. Limitations

There are several limitations in the research, although a clear contribution to the analysis of social impact is apparent. First of all, the number of participants was limited to 62 responses. In this sense, the results of the study cannot be generalised to a larger population. 

## 4. Conclusions

The hGLUTEN tool has an emerging social impact. The research and development process is still in its initial stages, with the execution of the first two prototypes. The tool is expected to be marketed in the near future with the collaboration of other research groups and food industry firms that may inject liquidity in the project. Co-creation activities have formed part of every stage of the process to improve users’ quality of life, in this case, persons who are allergic and intolerant to gluten and spoiled food.

The hGLUTEN tool has awakened users, engineers and professional chefs to the positive idea of its usefulness and necessity (Q1P, Q7U). They believe that it can be used to prevent many risk situations caused by ingestion of certain foods, which will enable closer and more personalised monitoring (Q5P, Q11U). This improvement is aligned with the social impact indicators (Addressing global challenges) established for the European Commission. Accordingly, it is a short-term improvement indicator in line with some of the targets set out in SGDs 3, 9, 10 and 11. 

It would be wrong to think that the use of this tool would not involve considerable changes in professional chefs’ work practices. This seems to indicate that this group is not capable of visualising the changes that the tool could bring to their daily work. They would therefore have to be guided in this aspect. Only 47% believe they could create new dishes and offer customers more convenience, security and service (Q4P).

Applied research pursues the production of new scientific knowledge with a business objective. It seeks practical solutions to problems that arise in invention of products that could be patented. An invention is the first creation of knowledge [47]. As regards the knowledge created by this device, there would be a clear impact on the entire food services industry (cafés, cafeterias, restaurants, catering, school dining rooms, hotels fast food, etc.) This impact could also be noticed in users’ homes. This would involve achieving the medium-term indicators of addressing global challenges and short-term indicators of R&I missions.

Food allergies rank fourth among the main public health problems according to the World Health Organisation (WHO). They affect 6 to 8% of children and 2.4% of adults [48]. Taking this into account, we can determine the scope of efficacy that large-scale replicability of this device would mean. A large majority of the professionals surveyed (41% of engineers agree-and 53% totally agree, 53% of professional chefs agree, and 26% totally agree) would be in favour of adapting the detector to other types of allergens (Q7P). This indicator would comply with one of the social impact criteria found in SIOR, the short-term indicators of Engaging EU Citizens.

The co-creation process was highly considered, and potential users were asked to take part in development of the device and make suggestions. All of the actors involved were very receptive. A total of 85% of users, 100% of engineers and 68% of professional chefs believe it is very important to give their opinion, which shows a growing level of involvement among the general population (Q8P, Q4U). Science is increasingly important in our worldview and closer than ever to society. Our appreciation of science and scientists is particularly evident in today’s exceptional situation. We are grateful for the different vaccines that have been designed to fight COVID-19. In this case, and in the short term (indicators of Engaging EU Citizens and R&I missions), the necessary collaboration between society and scientists will help to improve future hGLUTEN prototypes and will also bring technology transfer and adoption of new knowledge.

One limitation of this study is that the knowledge of this tool has not been shared with more people, as certain weaknesses in this aspect were detected during the research stage (Q3P, Q13U). This means that emphasis must be placed on dissemination in the media to increase its impact. The economic impact of an innovation largely depends on the dissemination process. Not all innovations are successfully disseminated [49], and therefore, this aspect should be highlighted. The dissemination of the results of a project is the first step to achieving social impact, and the next is transfer. Once people have found out the results of research, they must use them [25].

This study has demonstrated that this is a simple, fast, efficient and user-friendly application (Q1U, Q5U, Q7U, Q9U, Q14U). In this paper, the social impact indicators have focused on making progress towards the following Sustainable Development Goals 3—Health and well-being, 9—Industry, innovation and infrastructure, 10—Reducing inequality and 11—Sustainable cities and communities. Funding has already been obtained from the Basque Government for the TECAM project that analyses the presence of pathogens in dairy products and intolerance to gluten and soy with cross-validation of NIR to traditional PCR. In the long term, efforts will made to improve the tool’s artificial intelligence, and the impact indicators for the new version will again be verified. Once this step has been completed, the commercialisation stage is expected to begin.

## Figures and Tables

**Figure 1 ijerph-18-12722-f001:**
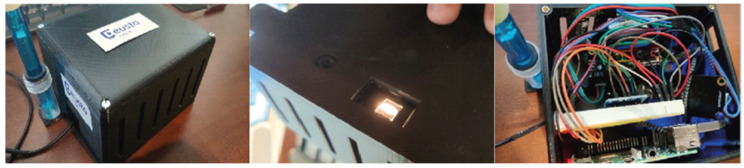
hGLUTEN device, exterior and interior.

**Figure 2 ijerph-18-12722-f002:**
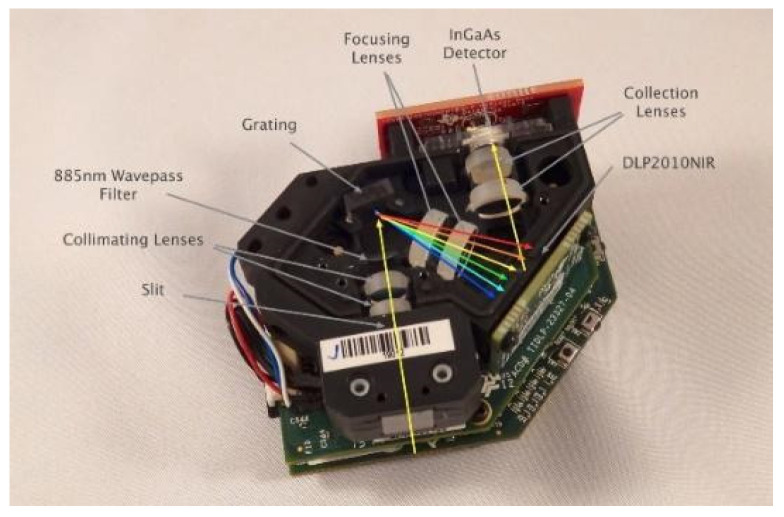
DLP NIR Nano Scan Source: Texas Instruments.

**Figure 3 ijerph-18-12722-f003:**
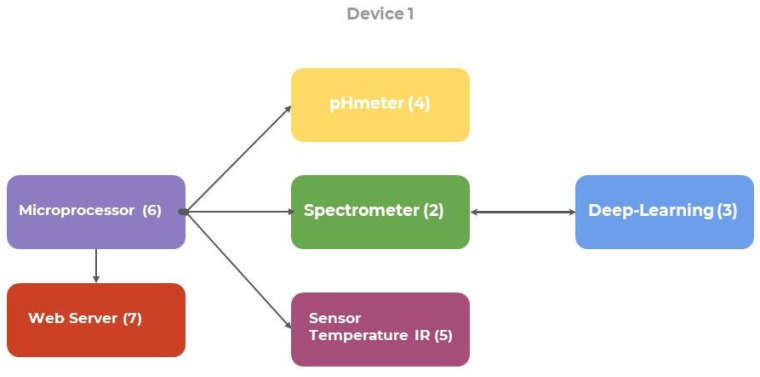
Diagram of system blocks.

**Figure 4 ijerph-18-12722-f004:**
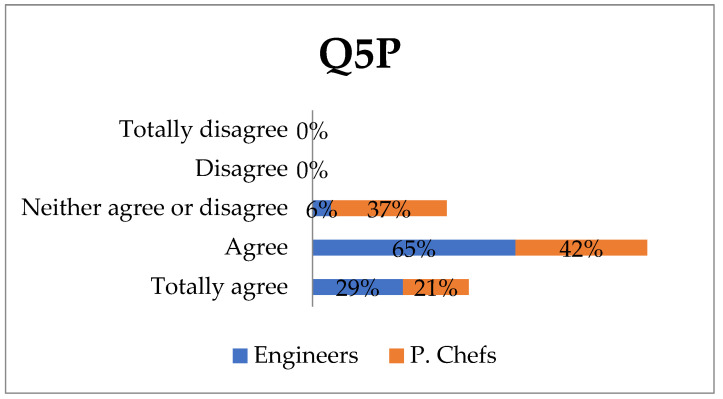
Change indicator of improvement in work practices.

**Figure 5 ijerph-18-12722-f005:**
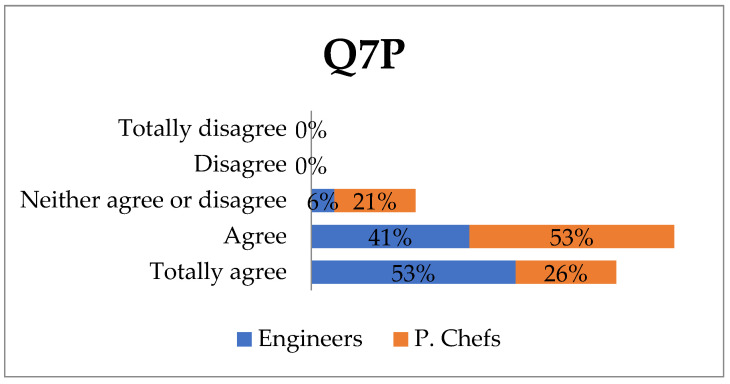
Indicator of evaluation and impact on professionals.

**Table 1 ijerph-18-12722-t001:** Engineers’/ Professional chefs’ data.

	Questions	Mean	Variance
Q1P	The new application would help people allergic to gluten or spoiled food avoid anaphylactic incidents in spoiled.	0.97	0.60
Q2P	Authorities could improve their policies and programmes with the data that the app provides.	1.11	0.44
Q3P	This information has been shared with family, friends, etc.	1.94	1.31
Q4P	The use of this tool could lead to changes in work practices.	0.33	0.23
Q5P	It would improve the detection of allergens and enable more personalised treatment than other tools that people know or use.	1.03	0.48
Q6P	I know about a type of scientific tool that enables people to detect allergens (gluten) or spoiled foods.	0.03	0.03
Q7P	The tool should be replicated for detection of other types of non-gluten allergens.	1.25	0.48
Q8P	We consider it necessary and important to give our opinion on research processes such as this one.	0.83	0.14
Q9P	Compared to other existing options, this new application could be used to customise data collection and processing.	0.67	0.23
Q10P	The hGLUTEN tool could be more involved in the treatment process.	1.06	0.28
Q11P	With this app, I could do a better job monitoring my customers’ allergy problems related to allergens and spoiled food.	1.11	0.39

**Table 2 ijerph-18-12722-t002:** Standard users’ data.

	Questions	Mean	Variance
Q1U	Patients with pathologies caused by an allergy to gluten, allergic reactions to spoiled food and even healthy people should have easy access to such services.	1.69	0.30
Q2U	This app would be effective for treating people with allergies to gluten or spoiled food.	3.31	0.86
Q3U	This app would prevent possible complications in pathologies caused by allergens that we are exposed to.	1.12	0.83
Q4U	In research processes such as this one, I consider it important to give my opinion.	0.85	0.14
Q5U	Access to data processing and the usefulness of tool portability.	3.08	0.87
Q6U	I would have instant access to the data whenever I needed it during the processing process.	3.08	0.87
Q7U	The tool’s interface must be easy to use to be effective.	1	0.72
Q8U	I know about another tool that people use to measure allergens in food or detect spoiled food.	0.12	0.11
Q9U	With hGLUTEN I could make as many measurements as I wanted or needed.	3.19	0.56
Q10U	This new app would increase the quality of life of people allergic to gluten and improve the service they receive in cafés, restaurants, schools, hospitals, etc., for example, by reducing travel time, leaving them more available time, using other alternative technologies or ways to save on other tools or treatments.	3.54	0.50
Q11U	Food services staff would monitor and handle risk situations more easily with this app.	1.08	0.63
Q12U	In order to understand how hGLUTEN works, the data provided by the engineers is important.	3	0.8
Q13U	I have commented and shared with others (friends, family, etc.) the knowledge and advantages that this application provides.	1.85	1.42
Q14U	If there was no medical or family support, the adaptation of this application would be good for the person using it in terms of complexity and knowledge of their disease.	1.36	0.24

**Table 3 ijerph-18-12722-t003:** Indicator of change in behaviour and practices.

	**Q1P**				
	**Totally agree**	**Agree**	**Neither agree nor disagree**	**Disagree**	**Totally disagree**
**Engineers**	30%	41%	29%	0%	0%
**P. Chefs**	21%	58%	16%	0%	5%
	**Q2P**				
	**Totally agree**	**Agree**	**Neither agree nor disagree**	**Disagree**	**Totally disagree**
**Engineers**	35%	59%	6%	0%	0%
**P. Chefs**	21%	53%	26%	0%	0%
	**Q3P**				
	**Nothing**	**Little**	**Not much or less**	**A lot of**	**Much**
**Engineers**	18%	24%	24%	28%	6%
**P. Chefs**	5%	26%	37%	21%	11%

**Table 4 ijerph-18-12722-t004:** Satisfaction indicator of follow-up and treatment.

	**Q10P**				
	**Totally agree**	**Agree**	**Neither agree nor disagree**	**Disagree**	**Totally disagree**
**Engineers**	18%	82%	0%	0%	0%
**P. Chefs**	16%	63%	21%	0%	0%
	**Q11P**				
	**Totally agree**	**Agree**	**Neither agree nor disagree**	**Disagree**	**Totally disagree**
**Engineers**	41%	53%	6%	0%	0%
**P. Chefs**	11%	68%	21%	0%	0%

**Table 5 ijerph-18-12722-t005:** Indicator of social improvement, health and quality of life.

**Standard Users**				
**Q1U**				
**Totally agree**	**Agree**	**Neither agree nor disagree**	**Disagree**	**Totally disagree**
73%	23%	4%	0%	0%
**Q2U**				
**Very effectively**	**Quite effectively**	**Effective**	**Little Effective**	**Nothing Effective**
58%	19%	19%	4%	0%
**Q3U**				
**Totally agree**	**Agree**	**Neither agree nor disagree**	**Disagree**	**Totally disagree**
42%	31%	23%	4%	0%

**Table 6 ijerph-18-12722-t006:** Indicators of improvements in the person’s experience.

**Standard Users**				
**Q5U**				
**Very useful**	**Quite useful**	**Useful**	**Little useful**	**Not useful**
38%	38%	15%	9%	0%
**Q6U**				
**Always**	**Often**	**Sometimes**	**Rarely**	**Never**
38%	38%	15%	9%	0%
**Q7U**				
**Totally agree**	**Agree**	**Neither agree nor disagree**	**Disagree**	**Totally disagree**
31%	42%	23%	4%	0%
**Q9U**				
**Always**	**Often**	**Sometimes**	**Rarely**	**Never**
39%	42%	19%	0%	0%

**Table 7 ijerph-18-12722-t007:** Indicators of improvement in services provided by health personnel for better monitoring and treatment of the pathology.

**Standard Users**				
**Q11U**				
**Totally agree**	**Agree**	**Neither agree nor disagree**	**Disagree**	**Totally disagree**
35%	38%	27%	0%	0%
**Q12U**				
**Very relevant**	**Quite relevant**	**Relevant**	**Little relevant**	**Nothing relevant**
31%	46%	15%	8%	0%
**Q13U**				
**Much**	**A lot of**	**Not much or less**	**Little**	**Nothing**
4%	31%	31%	15%	19%
**Q14U**				
**Totally agree**	**Agree**	**Neither agree nor disagree**	**Disagree**	**Totally disagree**
38%	62%	0%	0%	0%

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
