# Peer review of "hGLUTEN Tool: Measuring Its Social Impact Indicators"

_ijerph, 2021, doi:10.3390/ijerph182312722_

Round 1

Reviewer 1 Report

Thanks to the authors of the manuscript for research and development of a device that will help people with gluten intolerance and celiac disease to be safe about the products they choose on a daily basis.

Lines 149 and 156 (equation 1 and 2) - it is necessary to give explanations of the symbols used in the formulas.

2.1. Data collection.

Only 62 respondents took part in the survey conducted by the authors of the manuscript. More information is needed on the choice of respondents:

  1. What are engineers and professional chefs? What were the criteria for including these as the respondents? Do respondents deal with allergen management in companies?
  2. Very small number of standard user respondents? What were the selection criteria for standard user? Are they people with gluten intolerance or celiac disease or some other food allergy? The importance and necessity of the hGLUTEN device can best be assessed by people with any allergies or intolerances.

Lines 253-255 - Isn't there a mistake in the interpretation of the data?

2.2.2. Categorical data.

Lines 262-272, Tables 2 and 3 - very general information about the structure of the survey and questions for the respondents is given. At least once (eg Table 2 and Table 3) it is necessary to write which questions the respondents had to answer, not just the question number, for example Table 2 Q1P, Q2P, etc. If the survey questions aren’t given, then the idea that the authors of the manuscript wanted to say with the information given in Tables 2 and 3 isn’t perceptible.

Tables 4 and 5 - only one row in each table. This is not good scientific practice, so these two tables are not necessary. It is enough to explain in the text of the manuscript - Lines 269-272.

Lines 273-321 and Figures 4-13 - Text and information in the figures aren’t required as they are repeated / duplicated information with Table 1.

Results and discussion

Line 362 - conomic = economic

Figure 14, 15, 16 and 17 - there is no information in the graphs about which figure graphs applies to which question. Recommendation - combine the information in these figures into one table and then analyze all data together. Also, each figure has a graph, where there is only yes and no answers, which doesn’t need to be displayed in a separate graph, but analyzed only in the text.

Line 438 - is it in English?

Figures 18, 19, 20 and 21 - same comment and recommendation as for Figure 14, 15, 16 and 17.

Lines 521-526 - Did chefs and engineers give their opinion on the hGLUTEN tool after trying it?

There is no discussion or comparison with other studies conducted by other researchers who have studied the development and need for new equipment.

How many samples were tested to calibrate the hGLUTEN tool?

Lines 609-618 – Too general! How the short, medium and long-term missions will be achieved? What has already been done?

Lines 627-641 – Too general! In which countries an how many other European countries professionals are involved in the hGLUTEN development process and testing?

Lines 643-648 – The sentences are repeated twice!

Conclusions – Too general and some don’t follow from this study.

Author Response

Dear Editor,

  Thank you for the opportunity to revise our manuscript, hGLUTEN tool: measuring its social impact indicators. We appreciate the careful review and constructive suggestions. It is our belief that the manuscript is substantially improved after making the suggested edits.

Thank you for your consideration.  

Sincerely,

Reviewer 2 Report

The authors present a survey to determine the perception of the hGLUTEN tool for detecting gluten solids in foods.  Overall there is a positive reception of the technology, however it is unclear how much bias there may be in the populations participating in the survey which questions the validity of the results.  The paper also does not clearly describe any statistical methodology to determining significance of conclusions and hypothesis assessment.  For these reasons it is not recommended for publication, specific comments are as follows:

From the introduction it is unclear the background of the hGLUTEN device and publications, the first time it is mentioned in the final paragraph of Section 1 there is no citation.

Additionally, do the authors have any involvement or conflict of interest with the technology?

In the methods it is not clear how participants were recruited, how did they find the link?  It seems there would be a bias in a group participating in the survey to support and give their opinion in research projects which undermines one of the papers primary contributions in the abstract/conclusion for percentages willing to contribute.

For description of the interviewees was there a targeted demographic/age/sex, why were professional chefs and engineers and users chosen, this is not explained prior to Section 2.1.

Much of the plots from Figure 4 – 13 are redundant with Table 1 and could be removed, there is not meaning provided to why these results matter just a reporting of the numbers.  Is it needed to have equal men and women?  Why in Figure 11 are only two groups compared in the pie chart, what happened to standard users?

Table 2-9 it is not clear what questions are being asked and why Tables are organized and separated in these categories.

In Section 3 how was significance determined and what hypotheses were assessed to determine differences between populations?

Generally providing more context to questions within the results would improve readability, otherwise many of the plots are difficult to understand without having to search through the text for what specific question is being assessed.

Author Response

(The authors gave the same response as above.)

Round 2

Reviewer 1 Report

Thanks for the authors! 

The conclusions are too general.

Author Response

Dear Reviewer

We appreciate all your contributions to the improvement of this article. We intend to monitor the social impact of this tool in order to obtain more concrete conclusions in the future. We have also checked the spelling and grammar used.

Kind regards

Reviewer 2 Report

The authors have revised the manuscript and suitably addressed all concerns.

Author Response

Dear Reviewer

We appreciate all your contributions to the improvement of this article. We have checked the spelling and grammar used.

Kind regards